# A Multi-Level Feature Fusion Network for Remote Sensing Image Segmentation

**DOI:** 10.3390/s21041267

**Published:** 2021-02-10

**Authors:** Sijun Dong, Zhengchao Chen

**Affiliations:** Airborne Remote Sensing Center, Aerospace Information Research Institute, Chinese Academy of Sciences, Beijing 100094, China; jkobekawh@gmail.com

**Keywords:** remote sensing image, image semantic segmentation, scale difference, feature fusion

## Abstract

High-resolution remote sensing image segmentation is a mature application in many industrial-level image applications and it also has military and civil applications. The scene analysis needs to be automated as much as possible with high-resolution remote sensing images. This plays a significant role in environmental disaster monitoring, forestry industry, agricultural farming, urban planning, and road analysis. This study proposes a multi-level feature fusion network (MFNet) that can integrate the multi-level features in the backbone to obtain different types of image information. Finally, the experiments in this study demonstrate that the proposed network can achieve good segmentation results in the Vaihingen and Potsdam datasets. By aiming to achieve a large difference in the scale of the target objects in remote sensing images and achieving a poor recognition result for small objects, a multi-level feature fusion solution is proposed in this study. This investigation improves the recognition results of the remote sensing image segmentation to a certain extent.

## 1. Introduction

Semantic segmentation of the remote sensing images can be called pixel-level classification of the geomorphological features. By assigning a category to each pixel in the image, the location and semantic information are obtained. Due to the complex background and high density, most existing methods fail to accurately extract a road network that appears to be correct and complete [1]. With the development of deep learning, remote sensing image semantic segmentation that is based on deep learning has achieved great success. Since the first fully convolutional network was introduced, that proposed a fully convolutional neural network (CNN) that can solve the problem of image semantic segmentation, many classic image semantic segmentation algorithms have recently appeared [2]. In this study, the deep CNN method is used to extract and analyze the characteristics of the remote sensing images. From this, a multi-level feature fusion neural network is proposed in remote sensing segmentation to obtain the different features from the images, which can improve the results of the remote sensing image segmentation.

Remote sensing image segmentation differs from natural scene image segmentation. There are many natural scene categories, and different object categories often have strong interdependencies. For example, many objects will appear in a specific scene, which also leads us to using the image scene information to predict the object category. For example, sailing ships usually appear in the sea, and we can obtain the correlation of the sailing ships through the sea. However, remote sensing images are different from natural scenes. There are often not that many target categories that need to be identified, or only specific categories in remote sensing image scenes, such as buildings or roads, need to be identified. Therefore, we did not increase the interdependence of the image targets. At the same time, in the high-resolution remote sensing image scenes, there are often large differences in the size of the target objects. Therefore, our network must be robust to the difference in the target object scales. In this regard, multi-scale feature fusion is the focus of this experiment, which applies multi-scale feature fusion in remote sensing image segmentation [3]. Most CNN-based image segmentation methods mainly focus on deep or wide-area network architectures, while ignoring the correlation between the intermediate features [4,5]. In this study, we fused the high-level feature map with the semantic information and the low-level feature map with the image texture information so that the transfer of the feature information at the different levels is significantly enhanced, and more detailed feature information can be obtained. This study aims to build a reasonable structure for the remote sensing semantic segmentation model to obtain better segmentation results.

### 1.1. Related Works

Semantic segmentation algorithms are generally divided into the following categories: multi-level feature fusion [6,7,8], atrous convolution [9,10,11,12], UNet type networks [13,14,15], and boundary optimization [16,17,18]. The multi-level feature fusion method is well known to contain more spatial information in low-level feature maps, whereas high-level feature maps are richer in semantic information. Image semantic segmentation requires these two types of information to refine the result. As a result, some recent networks, such as RefineNet [19], PSPNet [20], and GCN [21], fused the feature map information at different levels.

Atrous convolution is another common method for optimizing the semantic segmentation algorithms, and this was proposed in Deeplab-v1. Through the use of atrous convolution, the convolution calculation can obtain a larger range of receptive fields, which is conducive to comprehensively extracting the target information by the convolution kernel. In addition, atrous convolution can expand the receptive field of the convolution without increasing the computation. In Deeplab-v2, Deeplab-v3, and Deeplab-v3+, the atrous spatial pyramid pooling (ASPP) module was proposed to combine the atrous convolution with the different atrous sizes to acquire receptive field information by using different scales. By using the above methods, the researchers optimized the results of the semantic segmentation algorithm.

Ronneberger et al. [13] conducted a study from a new perspective, in which the U-shaped network and biomedical image semantic segmentation are combined. The encoder in the UNet network structure is used to input the image and encode the image to extract the image information, and the size of the feature map is reduced by downsampling. In the decoder, the model upsamples the lower-resolution feature maps to the original image size to obtain the segmentation prediction. In addition, the skip connection is used to ensure that the feature maps in the encoding process of each stage are passed to the decoder. UNet mainly transfers the feature map of the encoding part to the decoding part, and it supplements the information of the encoding part in the decoding part; thus, the information in the decoding process is sufficient. UNet is a highly versatile network framework that has been used in the fields of remote sensing image segmentation and medical image segmentation. Another popular study is UNet++ [15]. In comparison to the original UNet network, UNet++ connects layers 1–4 that belong to UNet. The advantage of this structure is that the network combines all the deep features, which allows the network to learn the importance of the features for the different depths. The second advantage is that it shares a feature extractor, and it trains only one encoder, and its different levels of features are restored by the different decoder paths. The main improvement of UNet++ is to fill the original hollow U-Net. The advantage is that it can capture features at different levels and integrate them through feature superposition. Features of the different levels will have different receptive fields and they contain information about the different types and levels. For example, we can recognize large target objects more easily by feeling the feature map of the wild. However, in actual segmentation, the edge information for large objects and small objects is easily affected by the deep network. The calculation method of downsampling will lose some edge information. At this time, you may need to supplement low-level features in order to refine the edge information.

The boundary information in the images is also useful for semantic segmentation. Some methods for boundary information supplementation have been proposed. In addition, these methods concatenate different levels of features to refine the boundary. According to the different optimization methods that have been previously proposed, it can be observed that combining the features of the different levels is a ubiquitous optimization method. Our proposed MFNet (multi-level feature fusion network) strengthens the idea of feature fusion, and it adopts the method of combining the features of the adjacent levels to supplement the different scales of information that play a different role in remote sensing image semantic segmentation.

### 1.2. Contributions of MFNet

According to the network structure that is mentioned above, we can determine that the currently available segmentation algorithm improvement methods are mostly based on feature fusion. At the same time, for the remote sensing image segmentation, the size of the image target is not uniform and the network must be able to combine feature map information with different emphases. After analyzing the remote sensing dataset, we know that the category dependence in the remote sensing images is not very important. Moreover, there are fewer categories of remote sensing image datasets, and there is no definite relationship between the remote sensing image categories. Therefore, the interdependence between the categories is not necessarily useful for remote sensing image datasets [22]. As shown above, feature fusion is an important method in image semantic segmentation; therefore, we extended the feature fusion method to propose the MFNet.

The contributions of this study are as follows:(1)By focusing on the uneven size of the target object in the remote sensing image, a multi-level feature fusion network is proposed to extract the features, and it retains the information of the small objects as much as possible. In addition, it improves the recognition result of the small objects.(2)For the feature fusion module at the different levels, we proposed a cross-type feature fusion module, which can make the feature fusion module have effective semantic information and spatial texture information. In addition, it facilitates information exchange for the different features.(3)The proposed remote sensing image network model can have good target extraction and segmentation results in the urbanization remote sensing images. We significantly improved the segmentation results on the Vaihingen and Potsdam datasets.(4)The proposed multi-level feature fusion framework is a basic feature extraction framework that can be applied to the other remote sensing image deep learning tasks. At the same time, the cross-type feature fusion module is an embedding module that can be widely used to perform feature fusion.

## 2. Methodology

### 2.1. Multi-Level Feature Fusion Network

Most of the existing deep learning networks aim to deepen the network layer and increase the number of network structure parameters such that the network structure has better fitting capabilities. However, a considerable amount of information is lost in the process of network deepening because of the pooling module. In addition, this part of the information is difficult to restore in the decoder. Although some people supplemented the decoding module information by using methods such as SegNet [23] and UNet, the test results indicate that they still require further improvement. Naive information supplementation also interferes with the decoded information. Therefore, we require a network that can differentiate feature information that has been subjected to fusion. Based on the above ideas, this study proposes a feature fusion network to optimize the remote sensing image segmentation.

The multi-level adjacent feature fusion network mainly uses a multi-level feature map to enhance the fusion of information. For the input of the network, we chose 512 × 512 × 3 as the input size of the network. Level 1 to Level 4 in Figure 1 represent the network skeleton part of ResNet50. We designed a horizontally extended feature fusion skeleton network that is based on the four modules of the ResNet skeleton network. As shown in Figure 1, we used the four-stage feature map of ResNet as the initial framework for the expansion. The input image size was set to 512 × 512. The sizes of the feature maps for the four levels are 128 × 128, 64 × 64, 32 × 32, and 32 × 32. For the image features of the remote sensing images, we adjusted the downsampling of the initial skeleton network, so that the size of the last feature map was 32 × 32. Therefore, four-level feature map modules that have different information can be obtained and combined. We can define: function f as the convolution, g as a cross-type feature fusion module, and μ as the upsampling. Then, part of the encoding information extraction and restoration framework is shown in the following formulae:(1)$$l_{1}^{\prime }=g\left(l_{1},l_{2}\right)$$
(2)$$l_{2}^{\prime }=g\left(l_{2},l_{3}\right)$$
(3)$$l_{3}^{\prime }=g\left(l_{3},l_{4}\right)$$
(4)$$l_{4}^{\prime }=f\left(l_{4}\right)$$
(5)$$med1=g\left(l_{1}^{\prime },l_{2}^{\prime }\right)$$
(6)$$med2=g\left(l_{2}^{\prime },l_{3}^{\prime }\right)$$
(7)$$med3=g\left(l_{3}^{\prime },l_{4}^{\prime }\right)$$
(8)med4=g(med1, med2)
(9)med5=g(med3, med3)
(10)med6=g(med4, med5)
(11)output=μ(med6)

As shown in Figure 1, $l_{1},l_{2},l_{3},l_{4}$ represent the four different levels of the feature map of ResNet50. In addition, $l_{1}^{\prime }$, $l_{2}^{\prime }$, $l_{3}^{\prime }$,$\;l_{4}^{\prime }\;$ represent the results of the feature fusion, and mid1 to mid6 represent the information fusion intermediate layer feature that is generated by the cross-type feature fusion module in Figure 2.

Our semantic segmentation network reuses the features of multiple levels of the feature extraction skeleton network, which enhances the feature fusion and information exchange of the adjacent levels. After the previous description, image feature fusion is usually a common method that improves the results of the semantic segmentation. In general, the fusion of image features represents the supplement of the feature map information. For the image semantic segmentation projects, we need to downsample the feature maps to extract the semantic information and upsample to restore the image texture information. However, due to the existence of downsampling, the information is continuously lost, which is not conducive to the network in the process of upsampling to restore the target information of the original image. Therefore, we can optimize the decoding effect by continuously supplementing the information of the decoded module that belongs to the feature map. For the decoding module, the feature map information that is supplemented by the encoding part can be used to help understand the category information of the target in the image. In addition, the category information helps strengthen the recognition ability of the decoding module for the target category prediction. Therefore, we strengthened the feature fusion for the different levels of the feature maps and proposed a cross-type feature fusion method to optimize the feature fusion module. The number of categories for the remote sensing image datasets was less than the natural scene images. Therefore, it is easier for the network to extract semantic information from the remote sensing images.

### 2.2. Cross-Type Feature Fusion Module

By focusing on the feature fusion module, we also proposed a cross-type fusion module, as shown in Figure 2. It is usually difficult for us to effectively use low-level features; however, low-level features are an indispensable part of most computer vision projects. Generally speaking, the method of applying low-level features is used to connect them with the high-level features, which will result in a slight improvement in the prediction performance. An embedded feature fusion module was proposed to improve the use of the low-level features. This module fuses low-level features and high-level features as much as possible without sacrificing the spatial details of the former. Strengthening the information exchange of the different feature maps is conducive to fusing the features.

As demonstrated in the previous equation, MFNet uses the cross-type feature fusion module:(12)$$h_{1}=f\left(h\right)+f_{2}\left(l\right)$$
(13)$$l_{1}=\mu \left(h\right)+f\left(l\right)$$
(14)$$F=\rho \left(h_{1}\right)*l_{1}+l_{1}+\mu \left(h_{1}\right)$$
where  h is the high-level feature; l is the low-level feature; f is the convolution function and ReLU activation function; $f_{2}$ is the convolution function with a step size of two and the ReLU activation function; and $h_{1}$ and $l_{1}$ are the upper and lower layers, respectively. In addition, μ represents the upsampling by the bilinear interpolation, ρ is the change dimension of the global average pooling and the fully connected layer, and the symbol * indicates multiplication.

To distinguish the information between the different channels of the integrated features, because the semantic information of the deep features has a positive global information effect, a feature channel supervision module is added to the fusion feature map by using high-level features, and global average pooling is used to extract the global information. In addition, a fully connected layer is used to perform dimensional transformation to obtain the weight of the fusion feature channel [24]. The module uses this method to supervise the channel weighting on the low-level features. At the same time, to reinforce the high-level features and low-level features, the low-level features are connected by the residual method, and the high-level features with two updates are added and fused. Thus, a fusion feature map with comprehensive information on the adjacent levels was obtained. The experimental results show that the cross-type feature fusion module works well for the feature fusion.

This part is the cross-type feature fusion module that was proposed for the feature fusion. For the simple feature fusion methods, such as concatenation or addition, we did not effectively distinguish the information of the fusion feature map. For the semantic segmentation, the information represented by the feature maps at the different levels is different. To perform an effective segmentation, we usually merge the feature maps at the different levels together and then perform the decoding calculations. However, the fusion of the feature maps often requires adaptive learning of the features that have a different level of importance. For example, in most cases, the background information is not as important as the target information for the semantic segmentation algorithms. Therefore, we need a network structure to learn the differences between the different features. In the feature fusion part, a cross-type feature fusion module is added. On the one hand, the cross-type feature fusion method strengthens the information exchange and the fusion of the different levels for the feature maps. On the other hand, in the end, we adopted a simple channel weighting method for information screening and optimized the feature fusion method. In fact, according to the actual parameters of the network, we do not have too many parameters since we used 256 channels. Therefore, the calculation process is not particularly complicated when performing this part of the feature fusion calculation.

## 3. Results

### 3.1. Dataset

#### 3.1.1. Vaihingen Dataset and Potsdam Dataset Introduction

We set up experiments on the Vaihingen and Potsdam datasets and compared MFNet with some classic image semantic segmentation networks such as UNet, SegNet, PSPNet, and Deeplabv3+. The Vaihingen dataset specifically involved 33 high-resolution images. The main feature is the three-band data composition. Each image has a different size. It involves 16 real label graphics, which were divided into six categories. At the same time, in the aerial remote sensing data, the remote sensing image is taken from the top at an oblique angle or directly from the nadir. Due to the limited field of view, it may be challenging to distinguish the objects and materials on the ground only by looking at the appearance. For example, some flat roofs may have similar colors and spatial shapes as impervious surfaces. To meet this challenge, data collectors use a different sensing method that can measure supplementary information that can be used to distinguish ground objects with similar color characteristics. For example, the Vaihingen and Potsdam datasets use near-infrared, red, and green datasets [25]. The three bands are not common RGB images. Potsdam is similar to the Vaihingen dataset, and we used the same data processing method as the Vaihingen dataset for the experiment. We divided the 33 pieces of data from the Vaihingen dataset into 16 training sets and 17 test sets. During the experiment, we divided the 16 training sets into four images as the validation set.

#### 3.1.2. Dataset Crop Method

In remote sensing image segmentation, high-resolution remote sensing images are usually cut according to certain rules. We used 256 pixels as the overlapping area to cut the dataset into a 512 × 512 size as the training dataset, as shown in Figure 3. For the Vaihingen dataset, after completing the cutting and mirroring, we obtained 1324 images; however, we still need to focus on data augmentation during training.

#### 3.1.3. Data Augmentation Method

We used the following methods for data augmentation: (1) Flip transform: flip the image horizontally or vertically; (2) Random rotation transformation: randomly rotate a certain angle of the image; (3) Size scaling transformation: adjust the image according to a certain ratio; (4) Up, down, left, and right translation transformation: translate the image from the horizontal or vertical direction; (5) Random crop: randomly select a part of the image for cropping to obtain a partial image; (6) Contrast conversion: adjust the contrast according to the image.

### 3.2. Experiment Evaluation Metrics

The main evaluation method of the experiment is from the common semantic segmentation evaluations, which are variations in the pixel accuracy and region intersection over the union and F1 score [4], which is defined as follows. Assuming that there must be k+1 categories (including k target categories and one background category), this represents the total number of pixels that belong to category i but are predicted to belong to category j. Specifically, $p_{ii}$ represents the true positives, $p_{ij}$ indicates the false positives, and $p_{ji}$ denotes the false negatives:

Equation (15): Pixel accuracy:(15)$$pixel\;acc=\;\frac{\sum_{i\;=\;0}^{k}p_{ii}}{\sum_{i\;=\;0}^{k}\sum_{j\;=\;0}^{k}p_{ij}}$$
Equation (16): Intersection over union:(16)$$IoU\;=\;\overset{k}{\underset{i\;=\;0}{\sum }}\frac{p_{ii}}{\sum_{j\;=\;0}^{k}p_{ij}+\sum_{j\;=\;0}^{k}p_{ji}-p_{ii}}$$
Equation (17): F1-score:(17)$$precision\;=\;\frac{TP}{TP\;+\;FP}$$
(18)$$recall=\frac{TP}{TP+FN}$$
(19)$$1\;score=2\times \frac{precision*recall}{precision+recall}$$

### 3.3. Results and Visualization in the Vaihingen dataset

We conducted the experiments that are based on the Vaihingen dataset and compared MFNet with the classic image semantic segmentation networks such as UNet, SegNet, PSPNet, and Deeplabv3+. The same experimental conditions were selected during the experiment. For example, a data preprocessing method was carried out as mentioned above. Our hardware environment consisted of 2 NVIDIA Titan Xp, and we conducted comparative experiments on the different networks in this hardware environment. These experimental results in Table 1 show that the proposed MFNet exhibits a significant improvement.

To visually display our network results, we also performed plenty of visualization processing on the prediction results, as shown in Figure 4, Figure 5 and Figure 6. As shown in the local prediction results of the dataset, it can be seen that our method has a good segmentation result on the ISPRS Vaihingen for small targets such as cars in the yellow area and trees in the green area, and it will not lose the small target information. At the same time, the results for the low vegetables and trees that have nearly the same color also show a good distinction in the categories. This indicates that our feature map has good spatial texture information and semantic information.

As shown in the Figure 5 and the Table 2, we compared the visual results of MFNet with RefineNet, PSPNet, and Deeplabv3+. The comparison results are in line with our hypothesis. The extended network architectures such as RefineNet can effectively fit the edge of the scene, but it lacks semantic information, thus, making it difficult to distinguish the categories. PSPNet is relatively better at class classification, but this is because of the 8 times or 16 times downsampling that is used by PSPNet. It is impossible to effectively supplement the image detail information for remote sensing image segmentation. Therefore, when the model segmented small target objects, such as cars and small trees, the result was not that good. For the Deeplabv3+ network, the network aggregates the output of the ASPP with the low-level network to supplement the different levels of information, and then it performs upsampling four times to obtain the result. Compared with the other classical networks, it can effectively distinguish similar targets, and the segmentation results can effectively fit the ground truth. We propose MFNet to extract small targets and fitting the image edges, which strengthens the fusion of the different feature levels. MFNet was able to achieve very good results that can extract small targets in remote sensing image segmentation tasks. In addition, it solved the small targets problem during the remote sensing image segmentation tasks.

### 3.4. Results and Visualization in the Potsdam Dataset

For the Potsdam dataset, we also designed experiments to prove the effectiveness of the proposed network on the Potsdam RGB dataset, as shown in Table 3 Potsdam is a classic remote sensing image segmentation dataset for the ISPRS series. It consists of 38 high-resolution remote-sensing images. The image size of each image was 6000 × 6000, and the image sub-spatial resolution was 5 cm. Because the Potsdam dataset provides normal RGB three-band data, this experiment uses RGB three-band data, which is different from the Vaihingen dataset. According to the official requirements of the ISPRS, we divided 38 images into 24 for training, and the remaining 14 were used as the test sets. It contains the same categories as the Vaihingen dataset, and there are six categories each.

To verify the test and visualization effect of the Potsdam dataset, we performed a predictive visualization process on the pictures of the test set, and obtained the visualization results. This was done so that we have a clear and intuitive visual effect on the prediction results, as shown in Figure 7 and Figure 8. This is the prediction of MFNet in the Potsdam dataset. As shown in Figure 8, the MFNet that is proposed in this study has a good segmentation effect for large buildings and small cars. At the same time, the multi-level feature fusion network that was used in this study retains the complete image details. As a result, the segmentation edge is very close to the real picture. At the same time, the complete low-level image details enable us to extract the complete target objects, which includes small-sized target objects.

## 4. Discussion

### 4.1. Target Scale Difference

For the remote sensing images, there are usually a large number of targets with large-scale differences. For the same network model, it is usually difficult to extract targets with large differences. The calculation process of downsampling will lose information for the targets that have relatively small scales. Therefore, it is necessary to design a network that has better extraction capabilities for targets that have large-scale differences.

As shown in the Figure 9, the areas that are indicated by the light blue arrows represent low vegetation, and the areas that are indicated by the green arrows represent trees. It can be observed from the figure that the same type of target objects that are being pointed to with the same color have huge differences in the scale, and this includes small and large targets. In an image semantic segmentation network, targets with different scales usually require different levels of information for the identification and segmentation. Therefore, the network must combine the feature map information for the different levels to optimize the remote sensing image segmentation algorithm. The MFNet that is proposed in this study that is used to strengthen the multi-level feature fusion method can optimize the semantic segmentation of the remote sensing images.

### 4.2. Ablation Study in the Vaihingen Dataset

As shown in the Figure 10, the model that is proposed in this study has a better effect on Vaihingen, and the result after stitching has no obvious stitching traces. This indicates that the proposed model presents good remote sensing image segmentation predictions over a large scene range.

A multi-level feature fusion network is a commonly used optimization method in image semantic segmentation, and many previous studies are based on this. However, the multi-level feature fusion method that is proposed in this study is different from the previous methods. For example, SegNet and UNet adopt a feature fusion method that transfers the features of the encoding part to the decoding part. This increases the information of the decoding part, which makes the result of the decoding part more refined. However, this study adopts the feature fusion of the three levels for horizontal expansion, which strengthens the feature fusion. At the same time, the fusion method is different, and a cross-type fusion module is adopted. At the same time, RefineNet is also a classic stepped feature fusion easy segmentation network architecture; however, our structure is different from RefineNet. RefineNet acts on the semantic segmentation of the natural scenes, and the network goes from high-level semantic information to low-level semantic information. In the network structure that is proposed in this study, we use feature fusion from low-level detail information to high-level semantic information. We adjusted the downsampling multiple of the skeleton network so that the feature map belonging to the high-level part of the network has a larger size. Therefore, for the semantic segmentation of the remote sensing images, it is more appropriate to perform upsampling calculations. Therefore, we performed stepwise feature fusion that ranges from a low-level to a high-level based on the high-level feature maps. Finally, feature pyramid networks (FPN [33]) are also a common multi-level feature fusion algorithm framework. Our algorithm is different from FPN in that we adopted a stepped feature fusion method, which reduces the calculations. We do not need to obtain multiple levels of feature maps. Our goal is to aggregate the feature maps for the different levels on the high-level feature maps, and then we can decode them to obtain the prediction output for the semantic segmentation algorithm.

For the multi-level feature fusion network MFNet and the cross-type feature fusion module (CFM), we performed a comparative experiment on the Vaihingen dataset. The experimental results are presented in Table 4. As shown in Table 4, it can be observed that the original fully convolutional neural network that is based on ResNet50 has a preliminary segmentation effect on the Vaihingen remote sensing dataset. After applying multi-level feature fusion and connection, our network can compare the different categories on the Vaihingen dataset. As demonstrated, it has an excellent segmentation effect. As shown in Table 4, we repeatedly combined and used the features at different levels; thus, resulting in a 2.11% improvement in the accuracy of the building. It can be observed that our network strengthened the size of the remote sensing image scene. Objects with uneven targets were identified and effectively extracted. As shown in Figure 10, the visualization effect of our network after the CFM can effectively extract and identify the building area at the edge of the original image.

By focusing on the difficulty of recovering small target objects in remote sensing image segmentation tasks, we proposed an extended multi-level feature fusion segmentation network, MFNet, that is based on the RefineNet model. The MFNet model combines the features of multiple levels for cross-type feature fusion and it fully combines the information of the feature maps at different levels. Because the semantic information of the remote sensing images is simpler than the natural scenes, we chose feature fusion that is based on low-level networks. Therefore, we added a large amount of high-level semantic information to the low-level feature maps. This part of the information helps the network perform category recognition. At the same time, the feature fusion that is based on the low-level feature map helps us obtain the result by using four updates at the output of the network. Compared with eight times upsampling or 16 times upsampling, the four times upsampling result usually fits the target boundary, and the boundary texture information is more sufficient. The multi-level feature fusion skeleton network strengthens the fusion of the different content information so that the obtained predicted feature map can better retain detailed information, and at the same time, it can better predict the target object category. To strengthen the information exchange of the feature map fusion, we designed a cross-type feature fusion module to strengthen the information exchange and the fusion of the adjacent levels. Simultaneously, the channel supervision method was used to screen the features that are obtained by the fusion. The useful information is preserved and redundant information is suppressed; thus, we obtained a feature map with a high degree of information fusion.

## 5. Conclusions

By observing the image characteristics of the remote sensing images, we know that the size of the objects in such images is usually uneven. Large differences in the size of the same species are common, such as the low vegetation in the Vaihingen data. For semantic segmentation in the neural networks, due to the existence of downsampling, the information of the small targets in the image is usually lost. However, for semantic segmentation projects, this part of the information is usually required to supplement the detailed information. Therefore, in this study, we combined the information for the different levels in the image feature extraction network and combined the information of the low-level network to supplement the small target information that is lost during downsampling. In this study, we proposed a multi-level feature fusion network that is suitable for remote sensing image segmentation, and we achieved good results as demonstrated in the remote sensing image segmentation projects. Because the algorithm in this study is based on the feature extraction of the encoding part, it has good portability and can be easily embedded into the remote sensing image target detection algorithm. Moreover, as our algorithm retains the features of all the stages, there is no excessive loss of the details for small objects in the deep feature extraction process. Therefore, during the information restoration process, the comprehensiveness of the target detection can be enhanced. In addition, because this experiment only uses image information for three bands, the accuracy of the proposed algorithm can be improved by adding the fourth band information that belongs to the remote sensing images.

## Figures and Tables

**Figure 1 sensors-21-01267-f001:**
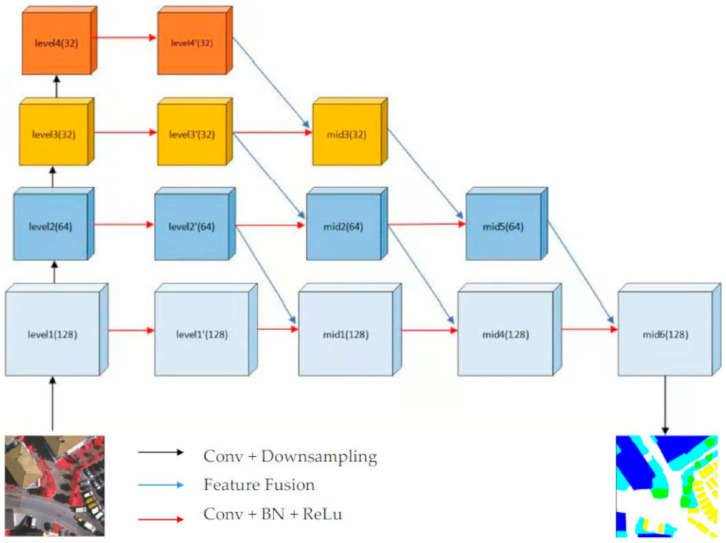
Architecture of MFNet.

**Figure 2 sensors-21-01267-f002:**
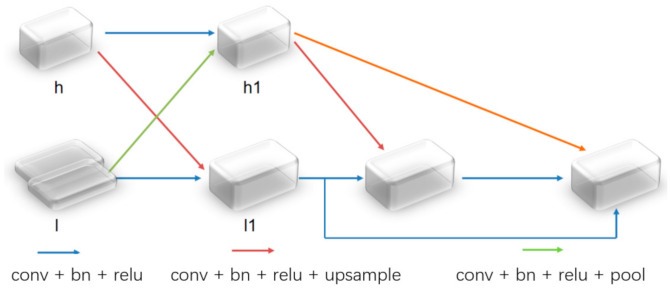
Cross type feature fusion module.

**Figure 3 sensors-21-01267-f003:**
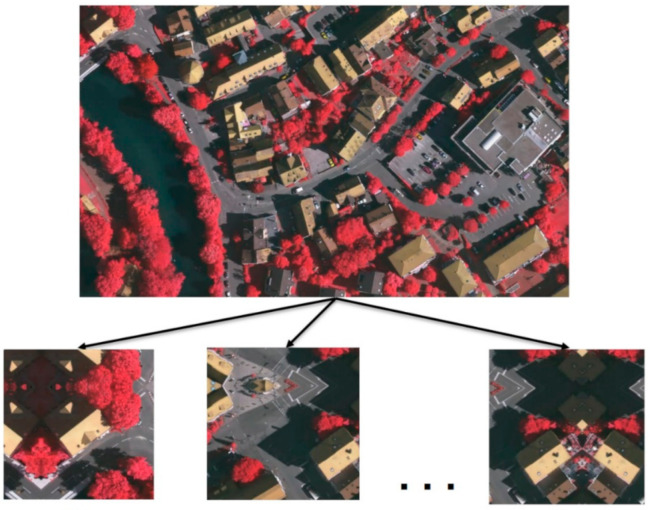
Data crop and augmentation.

**Figure 4 sensors-21-01267-f004:**
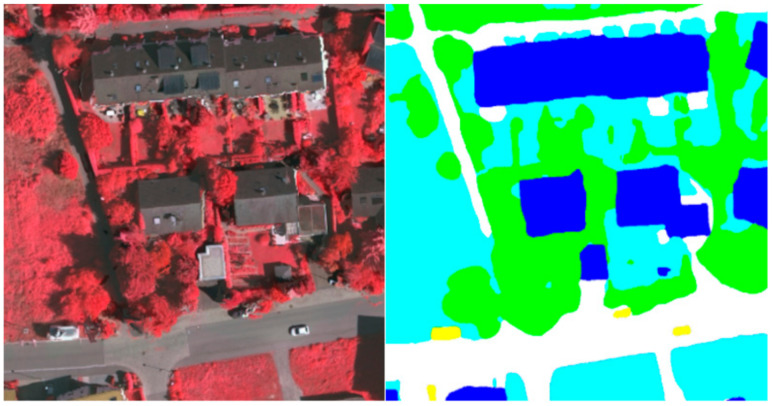
Visualization results of MFNet in the local area that belongs to the Vaihingen dataset.

**Figure 5 sensors-21-01267-f005:**
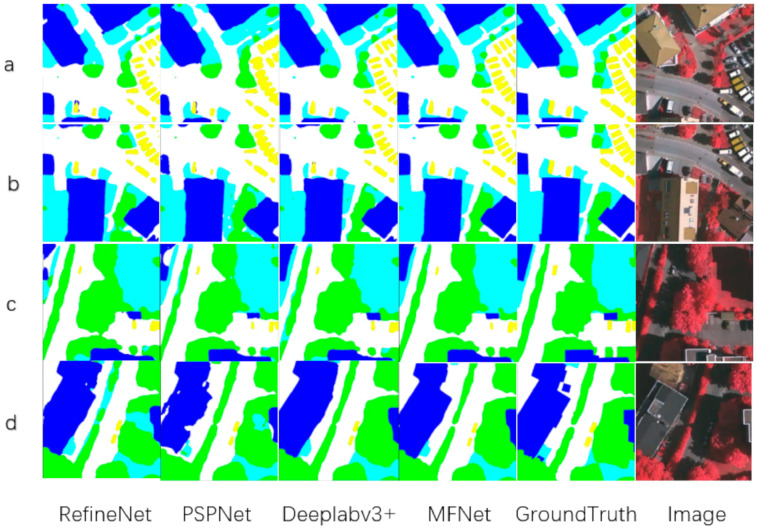
The result visualization of MFNet in Vaihingen dataset’s local area compare with other network (**a**–**d**).

**Figure 6 sensors-21-01267-f006:**
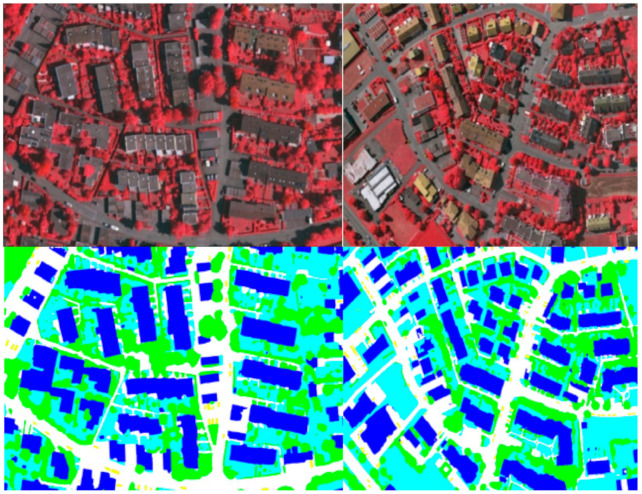
Visualization result of MFNet in a global area that belongs to the Vaihingen dataset.

**Figure 7 sensors-21-01267-f007:**
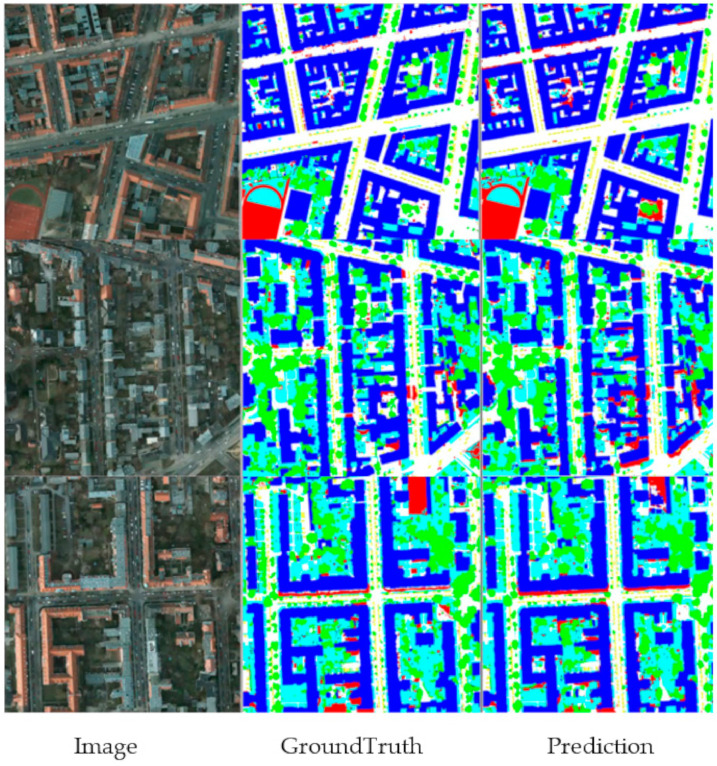
Visualization result of MFNet in the Potsdam dataset.

**Figure 8 sensors-21-01267-f008:**
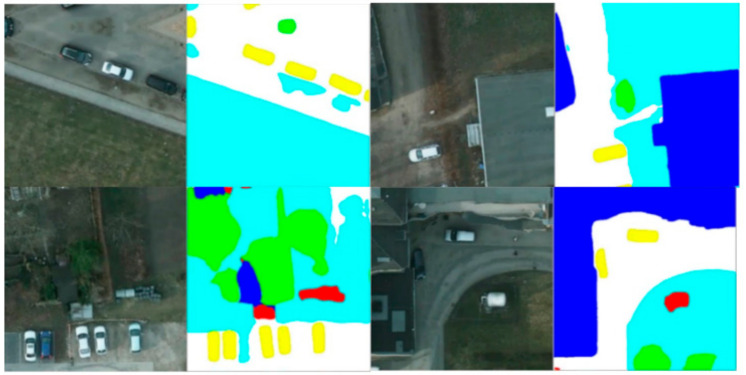
Visualization result of MFNet in the local area of the Potsdam dataset.

**Figure 9 sensors-21-01267-f009:**
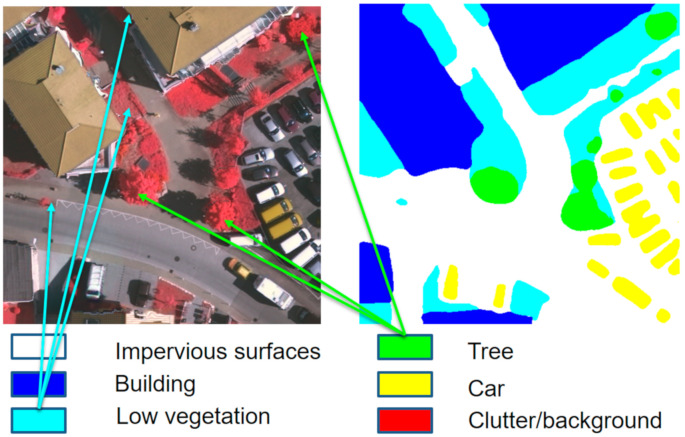
Scale difference in Vaihingen dataset.

**Figure 10 sensors-21-01267-f010:**
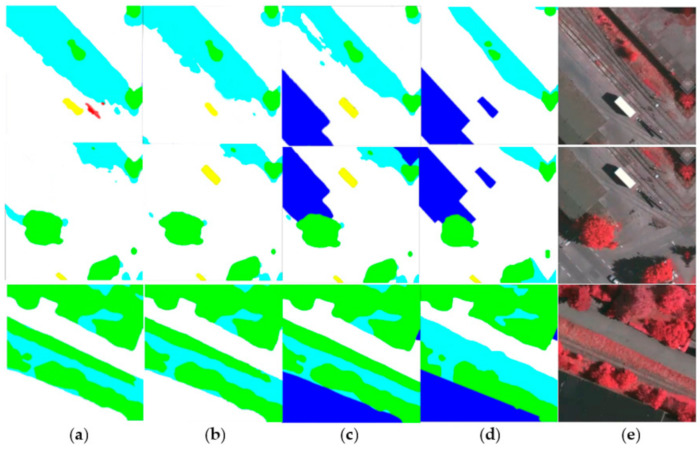
Visualization result of MFNet’s ablation study in the local area in the Vaihingen dataset: (**a**) Source FCN; (**b**) MFNet; (**c**) MFNet (CFM); (**d**) ground truth; and (**e**) the source image.

**Table 1 sensors-21-01267-t001:** Accuracy in the Vaihingen dataset (* defines the unknown backbone).

Method	Backbone	Accuracy (%)
FCN	VGG-16	82.39
SegNet	VGG-16	83.64
Edeeper-SCNN [26]	ResNet-50	85.78
SANet [27]	ResNet-101	86.47
EncNet	ResNet-101	86.63
RefineNet	ResNet-50	89.41
PSPNet	ResNet-50	89.50
UNet	ResNet-50	89.83
V-FuseNet [28]	*	90.04
Deeplabv3plus	ResNet-50	91.14
MFNet	ResNet-50 [29]	91.34

**Table 2 sensors-21-01267-t002:** Vaihingen dataset object accuracy.

Method	IoU (%)					Mean IoU	Mean F1	Overall Accuracy
	Imp. Surf.	Building	Low Veg.	Tree	Car			
RefineNet	79.47	86.05	66.03	76.24	58.65	73.29	85.57	89.41
PSPNet	82.14	87.12	66.69	74.57	56.87	73.48	85.82	89.50
Deeplabv3+	83.77	89.92	67.93	78.67	62.46	76.55	87.90	91.14
MFNet (CFM&DSM)	84.37	90.15	68.04	79.21	63.49	77.05	88.24	91.47

**Table 3 sensors-21-01267-t003:** Accuracy in the Potsdam dataset (* defines the unknown backbone).

Method	Backbone	Accuracy (%)
FCN	VGG-16	82.39
TreeUNet [30]	*	90.65
CCNet [31]	ResNet-50	90.87
HMANet [32]	*	92.21
MFNet	ResNet-50	91.65

**Table 4 sensors-21-01267-t004:** MFNet & source FCN & MFNet (CFM) accuracy in the Vaihingen dataset.

Method	Iou (%)					Mean Iou	Mean F1	Overall Acc.
	Imp. Surf.	Building	Low Veg.	Tree	Car			
Source FCN	77.28	84.25	63.81	74.46	57.97	71.55	83.65	87.12
MFNet	84.14	88.04	67.48	78.39	64.13	76.44	87.85	91.22
MFNet(CFM)	84.37	90.15	68.04	79.21	63.49	77.05	88.24	91.47

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
