# Peer review of "A Multi-Level Feature Fusion Network for Remote Sensing Image Segmentation"

_sensors, 2021, doi:10.3390/s21041267_

Round 1

Reviewer 1 Report

The paper focused on land cover classifications for high-resolution imageries using an improved deep-learning approach. As my understanding, the work is more of an improvement in the field of computer vision than an application in remote sensing. Below please find my major concerns:

  • The testing data sets are popular in semantic segmentation for high-res aerial images. It is remained unclear about how well the algorithm works when applying to another set of aerial or satellite imageries. For example, the work would be more significant if satellite images (e.g. those from Planet CubeSat or Chinese Gao-Fen) were involved.
  • The proposed algorithm did improve the classification performance but the gain is marginal compared with other approaches. Could the authors also list and compare the computational efficiency of these approaches?
  • Please briefly describe the computation environment (e.g. machine learning platform, libraries used, and hardware configurations).
  • Why not describe Potsdam data set in Section 3.1?
  • Please improve the writing. There are a number of grammar mistakes or repeated sentences to be taken care of. Just name a few:

        Line 26: “Semantic segmentation of remote sensing images is also called landcover classification” does not seem to be accurate.

        Sentences at lines 37 and 40-41 are largely repeated.

        Sentence at lines 67-78 is not clear.

         Line 335-336: “It is common to large differences in the size of the same species, such as …” The sentence is not clear.

Author Response

Dear reviewer:

      I think the advice given by the reviewer can point out my problem very well and is very helpful for me to revise the article. The problem raised by the reviewer is the missing or inaccurate description of my article. Thank you very much for your advice.Best wishes to you.

Sincerely.

Reviewer 2 Report

This paper present a technique on multi-level feature fusion network for remote sensing image segmentation. Some experimental results and analysis have been presented to support the work.

The novelty and contributions of this paper however, are very limited, if at all. The concept of multi-level feature fusion network is not new, some of which have been cited in the paper.

The fusion network (cross type feature fusion) as illustrated in Fig 2, looks very adhoc and trivial.

The results are rather mixed. On a number of cases, the proposed method does not outperform others. There should be a clear explanation of the reasons.

An ablative analysis re the various components/feature levels should be included.

The technique relies on fully labelled data, which might be costly. Rather weakly supervised approaches may be more appropriate for the task.

The paper reads heavy and occasionally it was hard to follow. Some typos and grammatical errors need to be fixed.

Author Response

Dear reviewer:

    I think the reviewer has made some good suggestions. For example, some modules need to be analyzed in more detail. At the same time, some experimental explanations should be made about the experimental results. These suggestions are very helpful to my article. Thank you very much for reviewing the reviewer.Best wishes to you.

Sincerely.

Round 2

Reviewer 1 Report

The authors responded my major concerns well, though a few minor revisions are needed:

(1) For the tests on Inria Aerial Image Labeling Dataset, can you provide accuracy metrics and compare with alternative algorithms? A brief summary of the comparison results is also recommended to be included in the paper. Such comparisons will help support your work.

(2) Please improve the writing. For example,  I had meant to rewrite the sentence (Line 385-386 “It is common to large differences ...") to be grammatically correct.

Author Response

Thanks for the advice, I have explained and supplemented all your suggestions, thank you for your review.

Reviewer 2 Report

I can see my previous comments are partially addressed but they are not adequate allayed. 

  • Still not convinced about the novelty and contributions of this paper. 
  • Point 3: the response is superficial. The revision alluded a number of terms eg scalability. there is no demonstration of it in the paper.
  • Meaning of "Not too many tracks are used, and Test Time Augmentation is not used in the testing method" is not clear. 
  • Point 4: The meaning of "By adding the channel weighting calculation method, the fusion feature map can be Learn the importance of
    different channel characteristics." is not clear. I guess it refers to the fact that the weight mechanism can be made learnable. It will be interesting to see that. 

Author Response

(The authors gave the same response as above.)
